# Major depressive episode externalizing symptoms among university students

**Betina Daniele Flesch**[ID]*, **Ana Laura Sica Cruzeiro Szortyka**[ID], **Gbènankpon Mathias Houvèssou, Fabiane Neitzke Höfs, Anaclaudia Gastal Fassa**

Faculty of Medicine, Federal University of Pelotas, Pelotas, Rio Grande do Sul, Brazil

* betinaflesch@gmail.com

## Abstract

### Introduction

Depression affects 32% of university students and Major Depressive Episode (MDE) externalizing symptoms indicate the severity of the case. This study assessed MDE externalizing symptom prevalence and associated factors among university students, with emphasis on aspects related to the academic environment, including interpersonal relationships and study area chosen by students.

### Methods

A census was performed on new students in the first semester of 2017 at a university in Southern Brazil. Depressive symptom prevalence was assessed by the Patient Health Questionnaire-9. The multivariate analysis of the associated factors followed a hierarchical model, using Poisson regression with backward selection.

### Results

MDE externalizing symptoms were present in 20% of the university students and were more frequent among females, those with brown, black or other skin color in comparison to those with white skin color. Individuals with a family history of depression, minority sexual orientation, negative self-reported academic performance, students taking courses in the area of Linguistics, Modern Languages and Arts and students who reported strong conflicts with university teachers or colleagues also had higher prevalence of externalizing symptoms.

### Conclusion

It is important for education institutions to train teachers to identify and deal adequately with students who have externalizing symptoms of MDE. In addition, education institutions need to pay attention to conflicts, both between students and also between students and teachers in order to take preventive measures in these circumstances.

**Data Availability Statement:** All relevant data are within the paper.

**Funding:** This research was carried out with the support of the Coordination for the Improvement of

Higher Education Personnel - Brazil (CAPES) -
Financing Code 001.

**Competing interests:** The authors declare that they
have no known competing financial interests or
personal relationships that could have appeared to
influence the work reported in this paper.

## Introduction

Young adults and adolescents, especially university students, face changes in the contexts of their lives, such as joining in new social groups and requirements related to the period of their academic and professional training. When these changes do not correspond to the expectations they and their families have created, emotional imbalance can occur which often serves as a trigger for suicidal ideation [1]. Studies indicate that between 15% and 25% of students develop some form of mental disorder when they are undergraduates, mainly depression [2]. With regard to major depressive episodes, externalizing symptoms stand out, such as the desire to injure/mutilate oneself and suicidal ideation.

Self-harm is classified as a self-inflicted physical injury behavior without suicidal intention, it is an indication that the individual is experiencing mental suffering; while suicidal ideation is the presence of thoughts, ideas and/or plans to commit suicide [3]. Mutilation, suicidal ideation, planning, attempts and suicide itself are symptoms related to depression. Little is known about depressive disorder externalizing factors (mutilation and ideation). However, it is known that they are particular phenomena, whereby ideation is more frequent among women and mutilation is typical of adolescents and young adults [4, 5].

Suicidal ideation prevalence has been measured in different ways among university students, varying mainly in relation to the recall period: whether ideation occurred during lifetime, in the last month or in the last two weeks. In Brazil assessment of a convenience sample of 637 university students in the state of Mato Grosso found 9.9% suicidal ideation prevalence in the last 30 days [6], while another study, with 515 adults resident in Campinas SP found 17.1% lifetime prevalence [7]. Studies in other countries point to even more alarming prevalence rates, such as 26% suicidal ideation prevalence among school students in Poland [8], and 9.7% in the last year among Mexican university students. Few studies have measured prevalence of self-harm among university students and those that have done so have found considerably diverging results, prevalence ranged between 0.8% and 5.3% in the last month among American and Mexican university students, respectively [9, 10].

Brazilian studies indicate that prevalence of suicidal ideation is higher among individuals with low income, who use alcohol and drugs, have minority sexual orientation and a family history of depression [6, 11]. Studies in other countries indicate that being of the female sex, following a non-Catholic religion or not having a religion, having minority sexual orientation and parents who are not married or not living together are risk factors for suicidal ideation and self-harm [9, 10].

Externalizing factors indicate the severity of depression and precede suicide, which is the fourth leading cause of death among young people aged 15–29 in Brazil [12]. As such, this study assessed prevalence and factors associated with externalizing symptoms of major depressive episodes among university students, with emphasis on aspects related to the academic environment, including interpersonal relations and the area of study chosen by students.

## Methods

This study is part of the research consortium "University Student Health", developed by the Post-Graduate Program in Epidemiology of the Federal University of Pelotas (UFPel). A census of UFPel undergraduate students who enrolled in 80 on-site courses in the first semester of 2017 and remained enrolled in the second semester of 2017 was carried out [13].

The total number of eligible students was 2706. Students who were under 18 years old or who suspended their enrolment during the course of the study were excluded.

Among the independent variables we assessed demographic aspects: biological sex (male or female), age (complete years, categorized into 18, 19–24 and $\geq$ 25 years old), skin color/race

(white, black, brown or other), self-reported family history of depression (yes or no)—considering family members to be those with whom the student lives every day or who are blood relatives—and predominant sexual orientation (heterosexual, bisexual, asexual or homosexual). We investigated socio-economic level, measuring consumer goods, and classifying according to the rating used by the Brazilian Association of Survey Companies [14] (A, B, C, D/E). Region of origin (Pelotas, other city in Rio Grande do Sul state, another Brazilian state or other country), and who respondents lived with (alone, parents or relatives, spouse, friends or colleagues) were assessed.

We also examined academic aspects, as knowledge area of the course (Exact and Earth/ Agrarian Sciences and Engineering, Health and Biological Sciences, Applied Social Sciences and Humanities, Linguistics, Modern Languages and Art), situation of strong conflict with teacher(s) (yes, but it didn't affect me, yes, it affected me or no), suffering verbal or physical aggression and/or having been humiliated by colleague(s) (yes, but it didn't affect me, yes, it affected me or no), as well as self-reported academic performance (bad/very poor, reasonable, good, very good/excellent). We also assessed the following behavioral variables: tobacco smoking (smoker, former smoker and non-smoker), harmful alcohol use according to AUDIT taking a cut-off point of $\geq 8$ points (yes or no) and consumption of illicit substances in the last month (yes or no).

Depressive symptoms were assessed using the Patient Health Questionnaire—9 (PHQ-9), which was chosen because it is an instrument that has been validated in the population of Pelotas [15] and because of its concise and easy to understand structure. The following question was used to measure major depressive episode externalizing symptoms: "In the last two weeks have you thought about injuring yourself in some way or that it would be better to be dead?" (yes/no).

Data collection was carried out between November 2017 and July 2018, by means of a self-administered anonymous questionnaire on tablets, using the Research Electronic Data Capture (RedCap) program, or printed questionnaires when the number of students exceeded the number of tablets available, or if the student preferred.

Master's Degree students recruited undergraduate students in their classrooms after obtaining the consent of the course coordinator and the teacher of the discipline in question. If undergraduate students were not found in the classroom, either because they were absent or because they were not enrolled in that discipline, they were recruited on another day, preferably in another discipline of the course. In order to achieve good quality data collection, a questionnaire manual was prepared foreseeing possible doubts. The Master's Degree students were trained to supervise the field work. Weekly meetings were also held with the aim of standardizing data collection and partial data were examined for consistency so as to identify potential problems early.

The data were analyzed using STATA 12.0. To characterize the sample, a descriptive analysis of the outcome and the independent variables was performed. Prevalence of major depressive episode externalizing symptoms was then calculated with its respective 95% confidence interval (95%CI). Associations between the independent variables and the outcome were calculated by prevalence ratios and respective 95% confidence intervals. The significance of associations was examined by the chi-square heterogeneity test and the linear trend test.

Hierarchical multivariate analysis was performed using Poisson regression with backward selection and adjustment for robust variance. The demographic and socio-economic variables were included on the first level. Region of origin, people they lived with, and academic aspects (knowledge area of the course, academic performance and disagreements with teachers and/or colleagues) were included on the second level, while the behavioral variables were included on the third level. Variables associated with the outcome with a p-value <0.20 were kept in the

model to adjust for confounding factors. Associations were examined by the chi-square heterogeneity test and the linear trend test considering those with a p-value <0.05 to be significant.

The study project was approved by the UFPel Faculty of Medicine Research Ethics Committee, Report No. 79250317.0.0000.5317 issued on October 2017. The subject of the research was explained to the students and their right not to take part and the confidentiality of the information provided was ensured. The questionnaires were only administered after students had signed the Informed Consent form.

## Results

A total of 1865 students were interviewed and 1849 answered the question: "In the last two weeks have you thought about injuring yourself in some way or that it would be better to be dead?", with a response rate of 68.3%. 55% of the population studied were female, 61% were between 19 and 24 years old, 72% reported white skin color, 56% reported family history of depression, 44% were of economic level B according to ABEP, 46% originated from Pelotas and 35% from other cities in Rio Grande do Sul state, 50% lived with parents and relatives, 40% self-reported their academic performance as good, 34.4% were taking courses in the areas of applied social sciences and humanities, 16% were smokers, 33% made abusive use of alcohol and 23% had used illicit substances in the last month (Table 1).

Out of all the students interviewed, 13.5% had suffered verbal or physical aggression and/or had been humiliated by faculty colleagues and 16.5% had had a strong conflict with faculty teachers. Self-harm behavior and suicidal ideation in the last two weeks were reported by 20.8% (95%CI 19.0–22.7) of the students (Table 1).

In the multivariable analysis, women presented 29% (95%CI 1.07–1.56) more externalizing symptoms than men. Individuals of brown and black or other skin colors had 40% more risk when compared to those of white skin color. Positive family history of depression increased by 36% the prevalence of major depressive episode (MDE) externalizing symptoms. Minority sexual orientation, i.e. homosexuality, bisexuality and asexuality, doubled the risk of developing externalizing symptoms, PR 2.26 (1.74–2.90), PR 2.57 (2.10–3.15) and PR 1.84 (1.28–2.64) respectively (Table 2).

The more negative self-reported academic performance, the greater the risk of developing symptoms, reaching risk of 2.6 when self-rating was "Very poor/Bad" (95%CI 1.91–3.56). Students taking courses in the area of Linguistics, Modern Languages and Art had externalizing symptom prevalence rates 53% (95%CI 1.19–2.01) higher than those taking Exact and Earth/Agrarian Sciences and Engineering. Students who suffered verbal or physical aggression and/or who were humiliated by faculty colleagues and felt affected by this experience had 64% (95%CI 1.32–2.04) more externalizing symptoms. Risk was 42% (95%CI 1.15–1.76) greater among students who had had strong conflicts with faculty teachers and felt affected by this. Tobacco smoking was also a risk factor for development of externalizing symptoms, being 52% greater among smokers (95%CI 1.19–1.94) and 28% greater among former smokers (95% CI 1.03–1.61) (Table 2).

## Discussion

MDE externalizing symptoms were found in 20% of the university students taking part in the study and were even more frequent among females, and in people with brown, black and other skin color than in people with of white skin color. Individuals with a family history of depression, with minority sexual orientation, negative self-rating of academic performance, students taking courses in the area of Linguistics, Modern Languages and Art and those who reported

**Table 1. Sample description according to demographic, socio-economic and academic characteristics and prevalence of externalizing symptoms of Major Depressive Episodes (MDE) among university students.** Pelotas (RS), Brazil, 2017 (n = 1847).

| Variables | N (%) | % Externalizing symptoms of MDE | p-value |
|---|---|---|---|
| **Sex** | | | 0.002 |
| Male | 831 (44.9) | 23.5 | |
| Female | 1016 (55.1) | 17.6 | |
| **Age** | | | 0.004 |
| 18 | 406 (22.1) | 18.2 | |
| 19–24 | 1116 (60.8) | 23.1 | |
| ≥ 25 | 314 (17.1) | 13.7 | |
| **Skin color/Race (self-reported)** | | | 0.004 |
| White | 1328 (71.9) | 18.9 | |
| Brown | 277 (15.0) | 25.6 | |
| Black or other | 242 (13.1) | 26.0 | |
| **Family history of depression** | | | 0.001 |
| No | 806 (43.7) | 17.3 | |
| Yes | 1040 (56.3) | 23.4 | |
| **Sexual orientation** | | | <0.001 |
| Heterosexual | 1381 (75.1) | 14.6 | |
| Homosexual | 144 (7.8) | 36.8 | |
| Bisexual | 237 (13.0) | 43.9 | |
| Asexual | 77 (4.2) | 28.6 | |
| **Socio-economic level** | | | |
| A | 264 (15.0) | 19.3 | 0.357* |
| B | 780 (44.2) | 19.1 | |
| C | 643 (36.4) | 21.9 | |
| D/E | 78 (4.4) | 25.6 | |
| **Region of origin** | | | 0.011 |
| Pelotas | 846 (45.8) | 19.5 | |
| Other city in Rio Grande do Sul state | 643 (34.8) | 19.3 | |
| Other state | 357 (19.3) | 26.6 | |
| **Lives with whom** | | | 0.001 |
| Alone | 231 (12.5) | 15.2 | |
| Parents or relatives | 929 (50.4) | 21.5 | |
| Spouse | 207 (11.2) | 14.0 | |
| Friend and colleagues | 478 (25.9) | 25.3 | |
| **Academic performance** | | | <0.001* |
| Very good / Excellent | 397 (21.5) | 17.6 | |
| Good | 739 (40.0) | 19.2 | |
| Reasonable | 609 (32.9) | 21.4 | |
| Very poor / Bad | 104 (5.6) | 41.4 | |
| **Courses (CNPQ main areas)** | | | <0.001 |
| Exact and earth/agrarian sciences and engineering | 539 (29.2) | 15.8 | |
| Health and biological sciences | 330 (17.9) | 16.4 | |
| Applied social sciences and humanities | 636 (34.4) | 22.6 | |
| Linguistics, modern languages and art | 344 (18.6) | 29.7 | |
| **Verbal or physical aggression and/or been humiliated by colleague(s)** | | | <0.001 |
| No | 1614 (87.0) | 18.5 | |
| Yes, but it didn't affect me | 83 (4.5) | 26.5 | |

*(Continued)*

**Table 1.** (Continued)

| Variables | N (%) | % Externalizing symptoms of MDE | p-value |
|---|---|---|---|
| Yes, it affected me | 157 (8.5) | 41.4 | |
| **Conflict with teacher(s)** | | | <0.001 |
| No | 1.546 (83.5) | 19.1 | |
| Yes, but it didn't affect me | 110 (6.0) | 18.2 | |
| Yes, it affected me | 196 (10.5) | 18.2 | |
| **Tobacco smoking** | | | <0.001 |
| Non-smoker | 1358(73.5) | 17.2 | |
| Former smoker | 202 (11.0) | 34.7 | |
| Smoker | 288 (15.6) | 28.5 | |
| **Abusive alcohol use** | | | <0.001 |
| No | 1133 (66.9) | 18.6 | |
| Yes | 561 (33.1) | 26.7 | |
| **Use of illicit substances (month)** | | | |
| No | 1423 (77.0) | 18.3 | <0.001 |
| Yes | 426 (23.0) | 29.3 | |
| **Major depressive episode externalizing symptoms** | | | |
| No | 1464 (79.2) | CI 77.3–81.0 | |
| Yes | 385 (20.8) | CI 19.0–22.7 | |

*Trend p-value

having had strong conflicts with teachers or colleagues at university had higher prevalence of externalizing symptoms.

MDE externalizing symptoms are important markers of severity, and self-aggression can leave irreversible physical sequelae. A publication on MDE prevalence and associated factors in this population already exists [16], but it was considered important to also explore the particularities of externalizing symptoms. This study stands out by investigating depressive episode externalizing symptoms in all undergraduate courses at a university, examining little studied aspects, such as association between conflicts in the university environment and these symptoms. Considering that the subject is delicate, the study used self-administered questionnaires, thus reducing information bias. However, a limitation of the study is the fact of not capturing separately information on suicidal ideation and self-aggression. This aspect, together with low availability of studies dealing with self-aggression, restricts comparability of the findings.

The literature available on suicidal ideation in university populations has reported very divergent results, whereby the sample of university students selected and the recall period need to be taken into consideration. Using a recall period of up to one month, the review study conducted by Pereira & Cardoso in 2015 found prevalence rates varying between 2.5% in American students and 14% in Norwegian students. When university students were asked whether at any time in their lives they had had suicidal ideation, prevalence became even more alarming, varying between 12.6% in a sample of Portuguese university students and 43% in Norwegian students [1]. Studies indicate 27.8% lifetime self-aggression prevalence and 12.2% (n = 4189) in the last 12 months among Mexican students [10], while a Brazilian study found lifetime prevalence of 70.5% (n = 17) [5].

Consistent with this study, the literature indicates that self-harm and suicidal ideation are more frequent among females, both in university students and in the general population,

**Table 2. Factors associated with externalizing symptoms of major depressive episodes among university students.** Pelotas (RS), Brazil, 2017.

| Variables | PR (95%CI) | p-value | PR (95%CI) | p-value |
|---|---|---|---|---|
| | **Crude Analysis** | | **Adjusted Analysis** | |
| **1st Level** | | | | |
| **Sex** | | 0.002 | | 0.006 |
| Male | Ref | | Ref | |
| Female | 1.33(1.11–1.61) | | 1.29 (1.07–1.56) | |
| **Skin color/Race (self-reported)** | | 0.004 | | 0.002 |
| White | Ref | | Ref | |
| Brown | 1.36 (1.08–1.71) | | 1.39 (1.11–1.75) | |
| Black or other | 1.38 (1.08–1.75) | | 1.40 (1.11–1.79) | |
| **Family history of depression** | | 0.001 | | 0.001 |
| No | Ref | | Ref | |
| Yes | 1.36 (1.13–1.64) | | 1.36 (1.13–1.64) | |
| **2nd Level** | | | | |
| **Sexual orientation** | | >0.001 | | <0.001 |
| Heterosexual | Ref | | Ref | |
| Homosexual | 2.52 (1.96–3.23) | | 2.26 (1.74–2.90) | |
| Bisexual | 3.00 (2.48–3.64) | | 2.57 (2.10–3.15) | |
| Asexual | 1.95 (1.34–2.84) | | 1.84 (1.28–2.64) | |
| **Academic performance** | | <0.001 | | <0.001* |
| Very good / Excellent | Ref | | Ref | |
| Good | 1.09 (0.84–1.41) | | 1.22 (0.96–1.57) | |
| Reasonable | 1.21 (0.93–1.57) | | 1.39 (1.07–1.79) | |
| Very poor / Bad | 2.34 (1.72–3.21) | | 2.60 (1.91–3.56) | |
| **Courses (CNPQ main areas)** | | <0.001 | | 0.002 |
| Exact and earth/agrarian sciences and engineering | Ref | | Ref | |
| Health and biological sciences | 1.04 (0.76–1.42) | | 0.98 (0.72–1.33) | |
| Applied social sciences and humanities | 1.44 (1.13–1.83) | | 1.28 (1.00–1.63) | |
| Linguistics, modern languages and art | 1.88 (1.46–2.42) | | 1.53 (1.19–2.01) | |
| **Verbal or physical aggression and/or been humiliated by colleague(s)** | | <0.001 | | <0.001 |
| No | Ref | | Ref | |
| Yes, but it didn't affect me | 1.43 (0.99–2.07) | | 1.27 (0.87–1.86) | |
| Yes, it affected me | 2.24 (1.81–2.76) | | 1.64 (1.32–2.04) | |
| **Conflict with teacher(s)** | | <0.001 | | 0.002 |
| No | Ref | | Ref | |
| Yes, but it didn't affect me | 0.95 (0.63–1.43) | | 0.82 (0.55–1.23) | |
| Yes, it affected me | 1.87 (1.51–2.31) | | 1.42 (1.15–1.76) | |
| **3rd Level** | | | | |
| **Tobacco smoking** | | <0.001 | | <0.001 |
| Non-smoker | Ref | | Ref | |
| Smoker | 2.02 (1.62–2.52) | | 1.52 (1.19–1.94) | |
| Former smoker | 1.66 (1.34–2.06) | | 1.28 (1.03–1.61) | |
| **Use of illicit substances (month)** | | <0.001 | | 0.092 |
| No | Ref | | Ref | |
| Yes | 1.61 (1.34–1.93) | | 1.30 (0.83–1.24) | |

*Trend p-value

although effectively committing suicide is more frequent among males [1, 17]. This fact may be related to females reporting depressive symptoms and recognizing them better and seeking more help with health problems than males [18]. Added to this is the difficulty men have in manifesting and recognizing emotions and feelings, which can result in psychosomatic symptoms. Women are more demanded in relation to academic performance and the care of family and friends and are more affected by relational difficulties, while men are more independent and very affected by financial difficulties [1].

Greater risk of externalizing symptoms among individuals of black and brown skin color is consistent with the literature, which points to association between these skin colors and the presence of major depression disorder symptoms. However, we did not find other studies about the relationship between skin color and externalizing symptoms. Individuals with non-white skin color tend to face greater obstacles, such as prejudice suffered in all realms of society and inside universities. In addition to this, there are great family expectations regarding their academic success, overburdening these students, who are often seen as responsible for improving their families' lives by becoming professionally qualified [18–20].

Studies indicate that higher drug use prevalence among university students is associated with depression and suicidal ideation. This association is subject to bidirectionality, given that drug consumption can be both a risk factor for unleashing depression symptoms, and also be a consequence, which encourages or is even a means of committing suicide [1]. Drug consumption is related to anguish and is used to escape from feelings or problems and can lead to exacerbation of depressive disorder externalizing symptoms [1, 6].

Consistent with the literature, this study found higher tobacco smoking prevalence among university students who have externalizing symptoms [1, 21, 22]. Tobacco consumption by university students is related to the desire to belong to the group, impulsive behaviors, quest for immediate pleasure and lack of perseverance. The habit of smoking serves as a way of relieving anguish and suffering [23, 24].

Occurrence of externalizing symptoms among university students with minority sexual orientation was greater. This is in agreement with the literature which indicates greater prevalence of depressive symptoms among homosexual and bisexual individuals. Such prevalence may be related to prejudice, lack of acceptance and stigmatization imposed by society, as well as family distancing and criticism and a smaller support network, leading to exacerbation of depressive symptoms and the appearance of self-injury and suicidal ideation [6, 25].

Positive association between presence of family history of depression and MDE is also in agreement with the literature [26, 27], whereby externalizing symptoms are part of depression. While some studies state that the relationship between family history and depression is genetic, others consider that it is more related to family environment [28]. In their systematic review, Bahls et al. indicate that family history of depression increases up to three times the probability of children and adolescents developing depression, and is the main risk factor [29]. Few studies with university students have assessed the relationship between the area of their courses and the presence of depressive symptoms, especially externalizing symptoms. This study found highest prevalence of externalizing symptoms in students taking courses in the areas of humanities, modern languages and art, similarly to the findings of another Brazilian study [30].

Association between depression and academic performance is described in the literature, and this study found that this relationship also exists with regard to externalizing symptoms, whereby academic performance is an important marker of success at university. This association is subject to bidirectionality, given that students who have MDE externalizing symptoms have greater difficulty in achieving academic success, but also low academic achievement can lead to depression. In addition, the greater criticality, characteristic of people with depression, may lead them to self-rate their performance in a more negative way [26, 28, 31].

Conflicts in the university environment were an important risk factor for the development of externalizing symptoms among university students. Interpersonal conflicts are important sources of stress in the general population. In the competitive university environment, such conflicts can be even more prejudicial. Conflicts between students can cause individuals to be isolated from their social group. Student conflicts with teachers can cause situations of chronic tension, with reduction in academic performance, change of classes or courses or even dropping out of university [32–34].

Studies suggest that prevalence of self-aggression is increasing, especially among younger people [9, 10]. Studies are therefore needed in order to gain more in-depth understanding of the particularities of externalizing symptoms in relation to major depressive episodes, analyzing suicidal ideation and self-aggression separately.

## Conclusion

Considering the important prevalence of major depressive episode externalizing symptoms among university students, education institutions need to be prepared to deal with these situations. Universities, in articulation with the National Health System, must ensure students health care, especially mental health care. University teachers need to be trained to identify and deal adequately with these situations. In addition, education institutions need to pay attention to conflicts, both between students and also between students and teachers in order to take preventive measures in these circumstances.

## Acknowledgments

I am grateful to all professors and colleagues in the Department of Social Medicine and the graduate program in Epidemiology at the Federal University of Pelotas (UFPel) and to all those who collaborated in any way in the elaboration and development of this article.

## Author Contributions

**Conceptualization:** Betina Daniele Flesch, Ana Laura Sica Cruzeiro Szortyka, Anaclaudia Gastal Fassa.

**Data curation:** Betina Daniele Flesch, Gbènankpon Mathias Houvèssou, Fabiane Neitzke Höfs.

**Formal analysis:** Betina Daniele Flesch, Ana Laura Sica Cruzeiro Szortyka, Gbènankpon Mathias Houvèssou, Fabiane Neitzke Höfs.

**Funding acquisition:** Betina Daniele Flesch.

**Investigation:** Betina Daniele Flesch.

**Methodology:** Betina Daniele Flesch, Fabiane Neitzke Höfs.

**Project administration:** Betina Daniele Flesch, Gbènankpon Mathias Houvèssou, Anaclaudia Gastal Fassa.

**Resources:** Betina Daniele Flesch.

**Software:** Betina Daniele Flesch.

**Supervision:** Ana Laura Sica Cruzeiro Szortyka, Anaclaudia Gastal Fassa.

**Visualization:** Betina Daniele Flesch.

**Writing – original draft:** Betina Daniele Flesch, Anaclaudia Gastal Fassa.

**Writing – review & editing:** Ana Laura Sica Cruzeiro Szortyka, Anaclaudia Gastal Fassa.

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
