## [Decision Letter · Decision Letter 0]

17 Mar 2021

PONE-D-20-37275

Major depressive episode externalizing symptoms among university students

PLOS ONE

Dear Dr. Flesch,

Thank you for submitting your manuscript to PLOS ONE. After careful consideration, we feel that it has merit but does not fully meet PLOS ONE’s publication criteria as it currently stands. Therefore, we invite you to submit a revised version of the manuscript that addresses the points raised during the review process.

Please be advised that submitting a revision does not guarantee acceptance.

We look forward to receiving your revised manuscript.

Kind regards,

Vincenzo De Luca

Academic Editor

PLOS ONE

Journal Requirements:

4. We note you have included tables to which you do not refer in the text of your manuscript. Please ensure that you refer to Tables 1 and 2 in your text; if accepted, production will need this reference to link the reader to each Table.

Reviewers' comments:

Reviewer's Responses to Questions

**Comments to the Author**

1. Is the manuscript technically sound, and do the data support the conclusions?

Reviewer #1: Yes

2. Has the statistical analysis been performed appropriately and rigorously? 

Reviewer #1: Yes

3. Have the authors made all data underlying the findings in their manuscript fully available?

Reviewer #1: Yes

4. Is the manuscript presented in an intelligible fashion and written in standard English?

Reviewer #1: Yes

5. Review Comments to the Author

Reviewer #1: This paper is a cross-sectional survey of University students in which they were asked about their experience of externalizing symptoms of depression and this was compared against a wide range of demographic factors. It is a purely correlational study so little can be said about causal relationships. It is unfortunate that after so much work there is not a lot of information actually about the relationship between different externalizing symptoms and these demographic factors. More nuanced questioning about those symptoms would have greatly enhanced the value of the paper. The authors do note this weakness. There are a numbers of typographic errors (for example in Table 1 they have the caption "Curses" instead of "Courses". A quick run through to improve the English would be good although for the most part it is reasonably well written. While I think the paper could be improved it is adequate for what it intends.

6. PLOS authors have the option to publish the peer review history of their article (what does this mean?). If published, this will include your full peer review and any attached files.

Reviewer #1: No

---

## [Author Response · Author response to Decision Letter 0]

28 Apr 2021

Dear Dr De Luca,

We appreciate the contribution of the reviewers to the article “PONE-D-20-37275 - Major depressive episode externalizing symptoms among university students”.

We have updated the article according to the requirements. Please find below the response letter addressing all the points raised and indicating the changes applied to the article. 

All the best,

Betina Flesch

Points raised and changes applied to the article:

The bibliographic references, line numbers, title page and file naming were reviewed to meet PLOS ONE’s style requirements.

The reference list was complete and correct, no papers were retracted.

The list of authors linked to an affiliation were included in a title page of the paper.

4. We note you have included tables to which you do not refer in the text of your manuscript. Please ensure that you refer to Tables 1 and 2 in your text; if accepted, production will need this reference to link the reader to each Table.

The Tables 1 and 2 were referred in the text.

Reviewer #1: This paper is a cross-sectional survey of University students in which they were asked about their experience of externalizing symptoms of depression and this was compared against a wide range of demographic factors. It is a purely correlational study so little can be said about causal relationships. It is unfortunate that after so much work there is not a lot of information actually about the relationship between different externalizing symptoms and these demographic factors. More nuanced questioning about those symptoms would have greatly enhanced the value of the paper. The authors do note this weakness. There are a number of typographic errors (for example in Table 1 they have the caption "Curses" instead of "Courses". A quick run through to improve the English would be good although for the most part it is reasonably well written. While I think the paper could be improved it is adequate for what it intends.

The paper was reviewed by a native English speaker and typographic errors were corrected.

---

## [Decision Letter · Decision Letter 1]

10 May 2021

Major depressive episode externalizing symptoms among university students

PONE-D-20-37275R1

Dear Dr. Flesch,

We’re pleased to inform you that your manuscript has been judged scientifically suitable for publication and will be formally accepted for publication once it meets all outstanding technical requirements.

Kind regards,

Vincenzo De Luca

Academic Editor

PLOS ONE

Additional Editor Comments (optional):

Reviewers' comments:

Reviewer's Responses to Questions

**Comments to the Author**

1. If the authors have adequately addressed your comments raised in a previous round of review and you feel that this manuscript is now acceptable for publication, you may indicate that here to bypass the “Comments to the Author” section, enter your conflict of interest statement in the “Confidential to Editor” section, and submit your "Accept" recommendation.

Reviewer #1: All comments have been addressed

2. Is the manuscript technically sound, and do the data support the conclusions?

Reviewer #1: Yes

3. Has the statistical analysis been performed appropriately and rigorously? 

Reviewer #1: Yes

4. Have the authors made all data underlying the findings in their manuscript fully available?

Reviewer #1: Yes

5. Is the manuscript presented in an intelligible fashion and written in standard English?

Reviewer #1: Yes

6. Review Comments to the Author

Reviewer #1: (No Response)

7. PLOS authors have the option to publish the peer review history of their article (what does this mean?). If published, this will include your full peer review and any attached files.

Reviewer #1: No

---

## [Editor Report · Acceptance letter]

1 Jun 2021

PONE-D-20-37275R1 

Major depressive episode externalizing symptoms among university students 

Dear Dr. Flesch:

I'm pleased to inform you that your manuscript has been deemed suitable for publication in PLOS ONE. Congratulations! Your manuscript is now with our production department. 

Kind regards, 

on behalf of

Dr. Vincenzo De Luca 

Academic Editor

PLOS ONE